# From Multidisciplinarity to Transdisciplinarity and from Local to Global Foci: Integrative Approaches to Systemic Resilience Based upon the Value of Life in the Context of Environmental and Gender Vulnerabilities with a Special Focus upon the Brazilian Amazon Biome

Anastasia Zabaniotou [1,*], Christine Syrgiannis [2], Daniela Gasperin [3], Arnoldo José de Hoyos Guevera [4], Ivani Fazenda [5] and Donald Huisingh [6]

[1] Biomass Group, Department of Chemical Engineering, Aristotle University, 54124 Thessaloniki, Greece
[2] Group of Studies and Research on Interdisciplinarity, Futures and Spirituality, Pontifical Catholic University of São Paulo, São Paulo 05014-901, Brazil; chrissyrgiannis@yahoo.com.br
[3] Group of Studies and Research on Futures, Pontifical Catholic University of São Paulo, São Paulo 05014-901, Brazil; palestrante@daniellagasperin.com.br
[4] School of Management, Pontifical Catholic University of São Paulo, São Paulo 05014-901, Brazil; arnoldodehoyos@yahoo.com.br
[5] Educational Program-Interdisciplinarity, Leader of the Group of Studies and Research on Interdisciplinarity—GEPI, Pontifical Catholic University of São Paulo, São Paulo 05014-901, Brazil; jfazenda@uol.com.br
[6] The Institute for a Secure and Sustainable Environment, University of Tennessee, Knoxville, TN 37996, USA; dhuisingh@utk.edu
* Correspondence: azampani@auth.gr; Tel.: +30-6945990604

**Abstract:** Economic and environmental interventions in the Anthropocene have created disruptions that are threatening the capacity of socio-ecological systems to recover from adversities and to be able to maintain key functions for preserving resilience. The authors of this paper underscore the benefits of a workshop-based methodology for developing a vision and an approach to the inner processes of creation that can be used to increase resilience, to cope with societal vulnerabilities and to develop the tools for future planning at local, regional and global scales. Diverse areas of discourse ranging from climate science and sustainability, to psychoanalysis, linguistics and eco-philosophy, contributed meaningfully to the transdisciplinary approach for enhancing resilience. A framework is proposed that can be used throughout society, that integrates the importance of human subjectivity and the variability of human contexts, especially gender, in shaping human experiences and responses to climate change impacts and challenges such as the covid-19 pandemic. Within the domain of socio-economic research, the authors challenge researchers and policy makers to expand future perspectives of resilience through the proposed systemic resilience vision. Movement towards transformative thinking and actions requires inner exploration and visualization of desirable futures for integrating ecological, social, cultural, ethical, and economic dimensions as agencies for catalyzing the transition to livable, sustainable, equitable, ethical, and resilient societies.

**Keywords:** resilience; climate changes; systemic; awareness; value of life; gender equality; amazon region; covid-19

## 1. Introduction

Resilience is acknowledged both explicitly and implicitly in a range of the targets of the United Nation's sustainable development goals (SDGs). The United Nations Agenda 2030 for Sustainable Development (SD) recognizes the importance of transforming societies through sustainable, resilient, and inclusive paths, encompassed by the seventeen interlinked and universal sustainable development goals (SDGs). For example, target 13.1 is focused upon strengthening resilience and adaptive capacities to climate-related hazards, to build the resilience of those in vulnerable situations and to reduce their exposure and vulnerability to climate-related extreme events and other economic, social, environmental shocks and disasters such is the covid-19 pandemic. This is the core feature of sustainable development, plus SDG5 was designed to help societies achieve gender equality and empowerment for all women and girls, which is a challenge for implementation of all SDGs [1].

Anthropocentric approaches and strategies of climate-based risk management are consequences of anthropocentrism [2]. The term "Anthropocene" refers to the human-dominated geological epoch, in which we are living [3]. In the Anthropocene, differential human vulnerability to environmental threats results from social, economic, historical, cultural, environmental, religious and political factors [4]. People living in fragile ecosystems are most at risk. Warming temperatures brought on by climate changes are enhancing the probabilities of new diseases to spread globally and to threaten societal resilience. Although urbanization, industrialization, and globalization have brought many benefits, their unsustainable implications have increased human vulnerability to some diseases, as illustrated by the current covid-19 pandemic that is severely disrupting the global community and economy with millions of infected people, and more than one million deaths as of 5 September 2020 [5].

Climate changes are shaping a new environment, which scientists predict will have dramatic anthropogenic impacts, which will create serious threats to societies, globally. Addressing these challenges and working on solutions for managing, reducing, and adapting to these risks are challenges that leaders of most countries have realized that must be "solved" to make progress toward sustainable and resilient present and future societies. To achieve reductions in risks from climate-change-related disasters will require societies to become more proactive, flexible, and resilient in integrated, multi-dimensional and inclusive ways [6].

Integration of preventative, mitigational, and adaptive policies, concepts and practices will require comprehensive approaches that should include close cooperation, synergy and coordination among policy makers, planners, institutions, local communities, and global society [7]. Furthermore, the holistic approach should be built upon transdisciplinary scientific collaboration that is not limited to technological achievements but is also based upon ecological, social, economic, and ethical dimensions [2]. Societies must be built upon gender, ethnic and racial equity in systemic ways that face the realities and threats of strengthening individual's and local communities' adaptive capacities. Currently this is largely missing in policy approaches and strategies [8]. Therefore, designing and implementing effective disaster, risk prevention, reduction and adaptation interventions requires attention to differentiated vulnerabilities and inequalities and social changes [4].

Many organizations, scientists and policy makers adopt strategies to manage risks, following two rationales for the management of social-ecological systems under uncertainty: (a) the controlling-the-risks rationale and (b) the resilience rationale. The controlling rationale has its roots in engineering and economics, with a strong history in computation and optimization. The resilience rationale is rooted in ecology, primarily focused on considering the diversity of the eco-social system's features [9]. More differences exist between the two rationales as is clear in the words used when referring to those two approaches. For the "*control*" rationale, words such as optimization, specialization, yield, performance, robustness, and control are used, while for the "*resilience*" rationale, words such as diversity, redundancy, vulnerability, coping capacity, flexibility, adaptation, adaptive management, and individual/societal transformation are used [9].

Planning of "*business as usual*" (traditional) strategies for adaptation often ignores the real factors that drive vulnerabilities and inequalities, because planners of traditional approaches put emphasis on

taking actions that serve to adapt a system to given circumstances and to provide practical guidance faced with immediate problems [10]. However, this is not enough for effectively facing the uncertainties of unprecedented risks associated with climate changes, covid-19-like pandemics, and other types of natural hazards, they require approaches that are built upon transformative thinking and actions [11].

Socio-ecological research findings warn us that there is an urgent need to seek alternative forms of thinking and action toward building sustainable and resilient societies [12]. The use of the word "resilience" for thinking about sustainable development, food security, health, adaptation to climate changes is very much on the research agenda, especially now with the covid-19 pandemic causing global health and economic disruptions that have increased human and societal vulnerabilities.

Issues such as building a dialogue between different scientific fields and incorporation of gender perspectives into development programs and environmental policies are urgently needed [12]. A recognition of the significant dimensions of gender in environmental sustainability and development is quite recent. Gender perspectives continue to be relatively underexplored by the scientific community of global environmental/technological research. It is important to address gender vulnerabilities because they provide another perspective for addressing global challenges, which can help to unify different disciplines within a single conceptual framework [13]. This opens a new academic perspective for building upon diverse fields such as ecology, complexity, change theory and systems theory, ethics, economy, artificial intelligence, food security, ecological refugees, dramatic climate changes far beyond 2 °C above the pre-industrial era average.

Until now, there has not been a clear framework of resilience that can be applied universally [14].

However, "the examination of resilience, and its relationship with vulnerability, reveals the need to tackle the philosophical questions that continue to blur the concept" [15].

*Scope and Objectives of the Study*

Coping with climate change-related risks is an opportunity of questioning our relationships with nature and ecosystems, and our values that drive inequalities in development and resist social reforms. Ensuring that people and communities are resilient to climate change is an urgent priority.

With the objective of contributing to systemic resilience thinking and action, the authors of this paper developed an approach/vision to enhance societal will and capacities to constructively contribute resilient thinking and acting by identifying current fatal flaws of societal behaviour in the Anthropocene. This was done by presenting findings of the benefits of a workshop-based visions and approaches to enhance resilience to climate change impacts, which are rooted in ecology, equality, and transformation, the insights emerged through activation of our inner processes of creation.

Moving from multidisciplinarity collaboration (multiple fields of knowledge not-connected) to interdisciplinarity (different fields of knowledge and disciplines inter-connected to help to catalyze global co-working to establish equitable and ecologically viable and resilient societies), and navigating across vulnerabilities and gender inequalities, from local to global scale, the authors documeted that diverse areas of discourse can contribute meaningfully to one another towards a transdisciplinary approach.

Multiple disciplinary approaches (multidisciplinarity) draw on knowledge from different disciplines to resolve real world problems, to create comprehensive research questions, to develop concensus environmental definitions and guidelines, but they stay within the boundaries of disciplines. Interdisciplinarity analyzes, synthesizes and harmonizes links among disciplines into a coordinated and coherent whole while transdisciplinarity integrates the natural, social and environmental sciences in a context of humanities and humanised care, and transcends the traditional boundaries of disciplines [16].

With the objective of developing a systemic resilience approach for shaping human experience and response to climate impacts and other challenges such as the covid-19 pandemic, the author's transdisciplinarity approach was based upon:

(1)    The epistemological systemic view of life.
(2)    The ontological awareness of inner processes of creation.

(3)    The social-epistemological gender vulnerability and equality frameworks.

This approach can be used locally, regionally and globally, in any context, because it can integrate the importance of human subjectivity and the variability of human contexts, especially gender.

By opening an interdisciplinary/transdisciplinary discourse on systemic resilience that can be applied to different contexts, the authors challenge researchers, scientists, engineers, policy-makers, decision-makers and individuals to expand their perspectives of resilience and transformation through the awareness of the value-of-life and our oneness with nature.

## 2. Methodology

To achieve the objectives of broadening the discourse of resilience perspectives in individuals and communities at risk, the authors used the following frameworks:

(i)     The Systemic View of Life (epistemological approach).
(ii)    The Value/Worth of Life (ontological approach).
(iii)   The Gender Equality principle (social epistemological approach).

The study includes an extensive and timely bibliographic search on resilience-related studies (From Capra and Luigi Luisi book, Freud, Lacan, Morin, Nicolescu, Fazenda, to Benyus studies, the Royal Swedish Academy of Sciences Special Issue on gender perspectives in resilience, vulnerability and adaptation to global environmental change, Center for International Forestry Research (CIFOR) studies, the Royal Society Publications, Philosophical Transactions of the Royal Society, working papers of international organizations including UN Women, United Nations Framework Convention on Climate Change (UNFCCC), Food and Agriculture Organization of the United Nations (FAO), European Parliament and Commission, open access publications most of the year 2020), and includes many insights from our previous publication [17].

The research on the individual's vulnerability and resilience was based on discourse analysis (DA), which works with language [18]. Discourse analysis (DA) works with language to give significance to the production of sense, as part of life. Taking in account that speech is language/communication/dialogue, the relationship between speech, language and ideology is discussed to unravel that behind speech there is an ideology, that is a set of ideas, thoughts, doctrines or world views of an individual [18]. Based on a systematic review according to the Cochrane Reviewers' Handbook [19], content analysis was performed, with exploratory and descriptive objectives. The databases used for systematic review included: (a) American Psychological Association (APA); (b) Science Direct and (c) Scopus (Scopus is Elsevier's abstract and citation database launched in 2004). In addition to searches in unpublished articles or grey literature, the following were used: (a) System for Information on Grey Literature (SIGLE), National Technical Information Service (NTIS), dissertation and theses database at Networked Digital Library of Theses and Dissertations (NDLTD) and conferences at SCOPUS. The evaluation of the quality of the studies was analyzed by three independent researchers, from the perspective of internal and external validity. Then, the analysis and interpretation of the data were based on the analysis of the content, according to Bardin (2009) [20]. The organization of the data content analysis was based upon three chronological segments: (1) pre-analysis; (2) exploitation of the material; and (3) interpretation of the results. To analyze the material, it was necessary to code it from the following procedures: (a) open coding designed to document the data and phenomena in the form of concepts; (b) the axial step focussed upon the improvements and differentiations of the categories that were revealed through the open coding; and (c) selective objective was to provide a brief descriptive overview of the history of the case.

The case study methodology was used with the Brazilian Amazon region as the focus. Case studies are employed in a variety of ways, depending to some extent on the specific subject area and geographic location, they have the potential to lead the research that is urgently needed. The author's case study was based upon literature search-based evidence and subsequent reflections that triggered the lines of discussion on systemic resilience, which was the most important part of the paper. It created arguments

and provided helpful nuances to evaluate and validate our approach/framework. We took the Brazilian Amazon biome as a case for reflections for the following reasons:

(i)     It is a domain of reflections for the Brazilian co-authors and others.

(ii)    It is a case for reflections on an eco-social system suffering from ecological vulnerabilities, social inequalities and unsustainable lock-in mechanisms (learning effects, economies of scale, economies of scope, technological interrelatedness, collective action, institutional learning effects and the differentiation of power) [21].

(iii)   It is a case where the authors could effectively articulate by using inputs from each of the disciplines used in the analysis. This was essential for grasping the full scope (community resilience and resilience of an individual who needs to be strong enough to overcome any adversity without failing).

Our discourse on resilience was focused upon two levels:

(a)    The individual's vulnerability and resilience.

(b)    The social-ecological system's vulnerability and resilience taking the example of the Amazon biome and gender vulnerabilities.

The authors proceeded in cascading steps, descending from the problem definition and objectives to the discussion of vulnerabilities and inequalities as explored in the Brazilian Amazon local scale. Then from the local scale, we worked to develop a conceptual framework of global applicability.

## 3. Case Study: Going Through Amazon's Vulnerabilities

Prior to proceeding with the reflections on vulnerabilities at a global scale and to develop our framework on systemic resilience, we considered that it would be focused upon associating the current socio-environmental characteristics of the Amazon region with climate change scenarios to identify those that may be most affected by climate changes. The Amazon region can provide valuable insights for reflection for the following reasons:

1.    It has been facing challenges inherent to the development projects in progress in the countries of the Amazon region, designed to promote social and economic progress with environmental conservation and climate change.

2.    In the field of Brazilian legislation, there was no improvement in any law to help to maintain the integrity of the Brazilian forests.

3.    Lock-in mechanisms based on the dominant ideology attract financial incentives from national and international investors for vegetable bio-extraction, thereby enabling big companies to remove invaluable raw materials to produce their products at no controlled rate [21].

4.    The ecosystem of the Amazon region is an important reservoir of biodiversity and local natural resources, which plays significant roles in serving essential functions in climatic balance with biospheric–atmospheric exchanges on regional and global scales [22].

5.    The Intergovernmental Panel on Climate Change (IPCC) focused upon the factors of exposure, sensitivity, and adaptive capacity in defining vulnerability as the predisposition to be adversely affected [23].

6.    Exposure, sensitivity, and adaptive capacity are influencing vulnerability according to characteristics inherent to the human or natural system of interest.

7.    In addition, understanding the geography of vulnerability to climate change results in more integrated disaster risk management, reducing the exposure of human and ecological assets and identification of particularly vulnerable populations [24].

A bibliographic search was performed of Amazon-related studies (regional studies particularly supporting broad claims, open access publications, CIFOR studies, Royal Society Publications,

Philosophical Transactions of the Royal Society, working papers of international organizations, European Parliament, etc.). Two ecologically related vulnerabilities were highlighted: (a) deforestation and (b) forest fires, both of which adversely affect the global climate and weather patterns, and contribute to the depletion of assets in medicine, agriculture, and other key industries, as a result of biodiversity regime loss in Amazonas [25]. A third vulnerability that is socially related (women vulnerability) was also highlighted. Finally, all these dimensions were interconnected. This interconnection reinforced the need for integrative measures in line with implementation of the SDGs.

*3.1. Moving Across Vulnerabilities*

The concept of vulnerability is rooted in hazard and disaster risk reduction and management as well as in studies on food security, poverty, and sustainable livelihoods [25].

3.1.1. Vulnerability One: Deforestation and Biodiversity Loss

Ongoing climate change is leading to an increase in frequency and magnitude of extreme climatic events in the Amazon which, in combination with other local human disturbances, is leading to unprecedented ecological disruptions. The Amazon region is the world's most biodiverse biome, corresponding to 53% of all tropical vegetation, including the North of Brazil, corresponding to 40% of all Brazilian territory, including the states of Pará, Amazon, Amapá, Acre, Rondônia, Roraima and some parts of the states of Maranhão, Tocantins, and Mato Grosso. It includes land from the Guianas, Suriname, Venezuela, Ecuador, Peru, and Bolívia; therefore, this region is a very important biome that must be preserved [26].

The Amazon basin encompasses 700 million hectares (M ha), and the central part lies entirely within the Brazilian territory [27,28]. Tropical deforestation in Brazil is a source of greenhouse gas emissions (GHGs) that contribute to climate change, having a direct impact on the weather and the humidity around the world. The Amazon forest is the reservoir of large quantities of carbon, both in plant biomass and in soil. Accumulated drought and deforestation have resulted in decreases in forest-based photosynthesis, thereby, undermining the Amazon region's role as a net capturer of carbon dioxide ($CO_2$), and its fundamental contribution to temperature, humidity, and rain pattern regulation [29].

Deforestation has accelerated the process of forest fragmentation in the Brazilian Amazon region, resulting in changes in carbon stocks in both biomass and soil [30]. The Amazon region has been undergoing profound transformations since the late 1970s, such as forest degradation, land-use changes, and effects of global climate changes [31]. In 1995, the deforestation of the Amazon forest had reached its peak, while a large decline in deforestation occurred in the Brazilian Amazon forest between 2004 and 2012. Forests store large amounts of carbon because trees and other plants absorb carbon dioxide from the atmosphere via the photosynthesis process. When forests are cleared or burned, the stored carbon is released into the atmosphere, mainly as carbon dioxide ($CO_2$). The result is that the Amazon forest faces climate change hazards [32]. During the years 2015–2017, the average global loss of tropical forests contributed about 4.8 billion tons of carbon dioxide per year to the atmosphere (or about 8%–10% of annual human emissions of carbon dioxide) [32]. In a devastating scenario, the Amazon basin's capacity as a "carbon sink" will drop to zero by 2030 [33].

Deforestation is also impacting the local biodiversity, which enables the system to function under a wide range of conditions and uncertainty. It impacts the bio–geo–chemical processes in forest ecosystems to an extent, depending on the relationships among the main components of terrestrial ecosystems, atmosphere, vegetation, and soil characteristics, water, and nutrient availability. Despite its biodiversity, the soil in the Amazon area is nutrient poor, meaning that, when the original vegetation is removed, there is no re-planting, because the soil cannot recover from deforestation [34].

However, Brazil has taken several steps to reduce emissions associated with deforestation, including the development of policies to promote and reward cattle ranching intensification [35]. Researchers argue that the intensification of shifting cultivation reduces the resilience of second-growth

forest; thereby, making this ancient form of land-use unsustainable [36]. In the field of agriculture and forestry, strategies typically are focused on a diversity of locally adapted crop varieties and animal breeds, crop rotation, intercropping, and extensive rather than intensive farming.

A new, integrated approach to multi-dimensional biodiversity is needed. A holistic, integrated approach is urgently needed to help achieve both climate and biodiversity global targets, involving:

(1)　Restoration and conservation of unprotected, degraded ecosystems.
(2)　Retaining the remaining strongholds of intactness.

To facilitate the implementation of global climate and biodiversity commitments at local levels, integrated high-resolution maps of carbon stocks and biodiversity that identify areas of potential co-benefits for climate change mitigation and biodiversity conservation are necessary [37]. In Brazil, there is no improvement in the field of legislation in any law to keep the integrity of the forest [38].

### 3.1.2. Vulnerability Two: Wildfires' Risk (Climate Change-Based Hazard)

Amazon forest faces climate wildfires' risk [39]. They are a major threat to the conservation of the Amazon forest because they cause huge carbon emissions, biodiversity losses, and local economic costs. During August 2019, about 76,000 fires burnt across the Brazilian Amazon forest [39]. Wildfires are anthropogenic and are due to the exogenous component of the increase in droughts [40]. However, a recent report reported that the fires in Amazonia are generally set intentionally and are large-scale forest fires that follow the illegal extraction of valuable timber to make way for cattle ranching and large-scale agriculture [39].

The second-worst year was 2017 for global tropical tree cover loss according to the Climate Council (CC), because 39 million acres (157,827 square kilometers) of tree cover was lost in total throughout 2017, with the three top countries for tropical tree cover loss being Brazil, the Democratic Republic of Congo and Indonesia, [41]. In addition, during August 2019, about 76,000 fires were burning across the Brazilian Amazon, an increase of over 80 per cent over the same period of 2018 [42]. For the first time in more than three decades, deforestation increased during the period from January to April 2020, while data released by the Brazilian National Spatial Research Institute (INPE), indicated a record 51% year-on-year (December to March) increase in deforestation in the Amazon, equivalent to about 796.08 km$^2$, in 2020 [29]. This resulted in a big loss of ecosystems services.

These issues are of international concern because Amazonia provides ecosystem services fundamentally connected to its genetic diversity and its wide expanse of forests, but at the present rate of deforestation in those regions, it will not contribute to provide the global climate benefits it has been providing.

These issues are complex and need to be assessed with objective criteria; for example, biodiversity depletion and climate change are often assessed based on non-objective criteria and are unforeseen in the current trade regime. Recent research on Brazilian Amazon forests indicates that isolation from direct deforestation or degradation may not be enough to maintain its ecological integrity in the future [42]. Systematic reforestation in this region and many other regions of the planet are urgently needed.

### 3.1.3. Vulnerability Three: Women Differentiated Vulnerability

While adaptation to climate change has been the dominant focus of many policy and research agendas, it is essential to ask why some communities and women are disproportionately exposed to and affected by climate threats [4].

Beyond biodiversity, social diversity in the Amazon system is fundamental to the capability of coping with climate change impacts and it refers to a variety of elements including species, people, strategies, behaviors, organizations, institutions. Approaches to managing climate change without acknowledging diverse socio-economic barriers to resilience can be socially exclusionary [43]. The perception of changes by local communities is important for risk analysis and for subsequent

societal decision-making. Strategies for community climate change coping must acknowledge, and be designed to correct, social factors/inequalities [43,44].

Indigenous communities can be important players in efforts to reduce forest carbon emissions because they have the knowledge and the consciousness of the forest lands [45]. Although researchers acknowledge local knowledge as a rich source of information, recognizing its potential to enrich the evidence basis for biodiversity conservation and sustainability, this is often neglected in the existing lock-in mechanisms and strategies [19,46,47].

However, enormous gaps in our knowledge about how gender relations shape the lives of people living in the forests of the Amazon exist. The theoretical content of gender as a concept is difficult to be applied to societies without an institutional framework of economic or political power, and functional work specialization [48]. Most scholars agree that in such societies, relationships between men and women can be defined in terms of complementariness, which does not necessarily entail a form of male power that involves the subordination of women [13]. In these contexts, such as in the Amazonas, the inequality experienced by women depends on its intersection with other forms of inequality, therefore gender inequality is understood within a framework of social inequalities. Additionally, the discussion should be framed on women and other marginalized groups being active agents for transforming and adapting to change, collectively [13].

In this study, the authors investigated women's differentiated vulnerability because the anthropogenic transformations of the Amazon region have increased the vulnerability of women. We must make this distinction because failing to consider gender vulnerability may lead to conservation initiatives and development interventions that do not reflect men's and women's respective views in the negotiation of trade-offs between different ecosystem services [49].

Women are more affected by climate change-based hazards, for the following reasons [38]:

(i)     They constitute most of the population.
(ii)    They constitute the majority of the poor.
(iii)   Their livelihood is more dependent on natural resources that are threatened by climate change.
(iv)   They face social, cultural, economic, and political barriers that limit their coping capacity.

Recent research conducted with nine indigenous communities in the Colombian Amazon, to understand which ecosystem services men and women perceive as most important for their wellbeing, revealed that there is an urgent need to incorporate a gender-based analysis in the assessment and valuation of ecosystem services in sustainable development projects because applying a gender lens would help society to understand which services contribute to their wellbeing. For example, efforts to promote the wellbeing of women could include improving the commercialization channels for locally made fruit juice, while efforts to promote the wellbeing of men could emphasize the sustainable management of fish populations [49].

Recent research documented that socioeconomic inequality influences the links between climate and security in complex ways. Women have far less access to spaces, resources, market opportunities, and political participation. The accelerating disruption of the Amazon rainforest has impacts on climate justice and human rights, particularly those of women and girls. The impacts of climate change (drought, floods, extreme weather, increased incidence of disease, and growing food and water insecurity) are affecting disproportionately the poor, the majority of whom are women. Since women's livelihoods rely heavily on local natural resources, climate change is making it more challenging for them to achieve and maintain water and food security while the decrease in those resources affects a woman's ability to care for her family and increases her workload. Furthermore, women's inadequate access to land makes them vulnerable to climate change impacts. Indigenous women are especially at risk, even as they engage in innovations in irrigation and agroforestry [50]. At the same time, foreign mining and development companies can have devastating consequences for the human rights of women and girls. For example, mercury poisoning from gold mining operations has caused

tremendous health challenges because it has polluted the rivers and the vegetation and has jeopardized Amazonian indigenous people [29].

Gender remains one of the least explored dimensions, despite indications that it is a key lens for understanding climate security risks, women are often discouraged from acquiring coping strategies in response to disasters [50]. Although women in the region suffer disproportionately from climate impacts, they also play an essential role in addressing climate change by contributing to both adaptation and mitigation efforts. They create innovative and localized solutions to build resilient communities. Indigenous women in Brazil are leading efforts to protest the destruction of their land and way of life [51]. Changes are happening across the region as women begin to organize and empower themselves. Networks of indigenous women in the Amazon have organized international meetings to find ways in which they can help forest conservation and food security for their communities and have been instrumental in making indigenous knowledge center stage [52].

The Center for International Forestry Research (CIFOR) was created to analyze the literature on gender and forests in Amazonia.In the very recent book of Colfer (2021) [53], forestry is considered as a masculine domain, and development focuses on the role of women. According to United Nations Development Program (UNDP) [54], there is a considerable gender gap in terms of political participation in Amazonian areas.

Although some government-led initiatives and legislation has been strengthening the rights of women to participate, this does not automatically provide women access to the political sphere. In general, to effectively incorporate women in the decision-making processes is a challenge and indigenous people's organizations are fighting to incorporate gender issues within their own internal processes [54]. Despite this, young indigenous women have recently emerged as climate leaders, being the organizers of the late 2019 climate strikes. This indicates that there are many opportunities for participation, engagement, and leadership for women [50].

Based on the above constructs and evidence, the authors of this paper emphasize that no change to non-gender sensitive processes in the Brazilian Amazon may result in the prioritization of objectives that do not include men's or women's perspectives, vulnerabilities and coping, which in turn may impact the Amazon system's capability for coping with climate change impacts. Gender vulnerability in natural disasters and adaptation needs to be studied for the Amazon region. A discourse on diversity/vulnerability needs to be opened and to focus upon the gender-differentiated vulnerability in natural disasters by using specific tools. Bringing gender mainstreaming into planning and policy development and designing and implementing adaptive strategies is also needed [44]. Finally, there is a need for a more grounded and localized understanding of global environmental change (GEC), which recognizes the experiences of individuals and communities bound in local places and local culture and knowledge [13].

### 3.2. Changing the Game in the Brazilian Amazon: Why a Change Is Needed?

There is a clear lack of commitment to nature conservation by the current administration in Brazil, with the use of command and control mechanisms in the environmental arena [28]. Although there is some resistance by congress, civil society, trade and economy sectors, the pattern of deforestation in Amazonia and other biomes is alarming. The current controlling approach focuses on managing the system's performance based on one or a few variables of short-term economic interest [55].

Brazilian researchers in 2018 mapped the vulnerabilities of the municipalities of the state of Amazonas using a climatic perspective, by developing a municipal vulnerability index, which was used to associate socioenvironmental characteristics of municipalities with climate change scenarios. These researchers found that from a socio-environmental and climatic point of view, these regions should be a priority for public policy efforts to reduce their vulnerability and to prepare them to cope with the adverse aspects of climate change [56]. These aspects of the climate, coupled with the dependence of the population of Amazonas on ecosystem services, constitute a scenario of a threat to human well-being, which can be aggravated by the poor social indicators of the region [56].

Nonetheless, nature protection and climate change mitigation and adaptation policies were developed separately, not only in Brazil but in many other nations. However, these separate efforts have failed to achieve the scale of action needed to decrease biodiversity losses and mitigate climate change [57]. In a study of the year 2020, five types of mechanisms that support the production of adaptation services were identified. Understanding these mechanisms can lead to an improved flow of adaptation services and more options for livelihoods and well-being under climate change [58]. These services are related to:

(i)    Multifunctional and traditional ecosystem management.
(ii)   Proactive management of transformed ecosystems.
(iii)  Use of novel adaptation services, collective ecosystem management, and appreciating, and valuing adaptation services.

The study on "Options for sustaining material and non-material benefits as ecological structure and functions transform", published in the Philosophical Transactions of the Royal Society Series B, advocates that understanding trade-offs and co-benefits of adaptation services (AS) is essential to support social adaptation [59].

"Nature-based solutions" (NbS) are proposed for contributing to protecting people from climate change impacts, supporting biodiversity, and securing ecosystem services [60]. According to some researchers, NbS benefits have not been rigorously assessed and there are concerns for their reliability and cost-effectiveness and their resilience to climate change is lacking. Furthermore, these researchers argued that NbS ignore the importance of complex ecological interactions across temporal and spatial scales. They suggested that the precondition for NbS to be effective in contributing to effective counter responses to climate changes and to biodiversity crises, is by, simultaneously contributing to sustainable development via systemic integration according to the tenets of the systemic change concept (SCC) [61].

To design new initiatives the context must be considered. To develop and implement initiatives with a greater chance of success, engagement must be integrated with proper context. The Center for International Forestry Research (CIFOR) has published many interesting studies for long term sustainability and resilience of forest and related communities. A CIFOR study advocates that for the long-term sustainability of the system, and due to biodiversity and environmental concerns, there is a need to see the context. There is a need to restore and utilize degraded land through an integrated production system that can also conserve the environment, supported by competent government policies and local communities with sufficient social capital [62]. Authors of another CIFOR study argued that restoration depends on purpose and context and entails innovation to stop or reverse the ongoing degradation (not necessarily recovering past system states) for increased functionality. Location-specific interventions to support and restore social-ecological systems need to go synergistically with the transforming drivers for reversing unsustainable land practices. The typology can help to link knowledge with action in people-centric restoration in which external stakeholders are seeking to admit shared responsibilities for historical degradation [63].

To shift from seeing context as an obstacle, multi-stakeholder forums (MSFs) are receiving attention due to the need to address climate change and transform development trajectories [64]. A study from CIFOR emphasized that the most successful multi-stakeholder forums are those that are recognized as part of a wider process that seeks to transform practices at multiple levels and are focused upon building consensus and commitment from higher levels, political will, and are designed as adaptive learning processes [65].

However, for the implementation of forest and landscape restoration (FLR) framework, existing guidelines and best-practice documents are not satisfactory. The single working framework is unlikely to be effective. Thus, there is a need to co-develop and apply specifically tailored working frameworks to help ensure that FLR interventions bring social, economic, and environmental benefits to multiple stakeholders within landscapes, being able to adjust to the uncertainty of changing conditions over time [66]. In other words, there is a need to take a people-centered land governance approach for the

Amazon region. Based upon research performed by the Center for International Forestry Research (CIFOR) and the Consortium of International Agricultural Research Centers (CGIAR), it was reported that because indigenous peoples' land and territorial rights are under threat, the International Land Coalition (ILC), the Global Land Program (GLP), and partner groups are committed to their recognition and protection, including recognizing respect to indigenous knowledge and cultures for contributing to sustainable and equitable development [67].

Both indigenous women and men depend on forests, agroforestry, and trees for their livelihoods, and play a critical role in managing them. In a previous study by CIFOR in 2016, it was suggested that adaptive collaborative management (ACM) should be applied, which is a social learning-based approach to help forest communities manage their natural resources in a more equitable and sustainable way. The ACM is designed to respond to change, to foster collaboration, and negotiation, and to build skills and capacities. It is also a sustainable way to promote gender equity (GE) among communities that are strongly patriarchal and characterized by cultural practices that exclude women from tree planting and land ownership [68]. The ACM team members and facilitators are necessary to address the complex nature of socio-ecological relationships and for building social learning processes.

Inequalities persist in roles, rights and responsibilities of women and men in communities with power relations. These inequalities shape the ways they participate in decision-making, how they benefit from forest and tree resources. Gender biases and exclusionary social norms in the wider policy result in a gender gap that concerns the access and control of key resources, including land, water, energy, food, labor, credit, information and extension services, with women facing disadvantages in several domains [69]. Addressing the gender gap in participation and representation in community forestry by action on gender and promoting women's participation in forest decision-making within their own rural communities requires consideration of gender relations and gender perspectives in the ACM [70].

Finally, management and conservation of the biological world (where we also belong) and eco-social systems requires a transition from trying to minimize biological change to facilitate dynamism to adjusting to change in the Anthropocene. Rapid anthropogenic climate change that is being experienced in the twenty-first century is intimately entwined with the health and functioning of the biosphere. In historical context, recent biological changes should be seen as responses to multiple drivers of change [71].

There is an urgent need to understand the ecological dynamics of climate impacts, to identify hotspots of vulnerability and resilience and to identify management interventions that may assist in supporting and enhancing biospheric resilience to climate change. Therefore, there is a need to adopt strategies of adaptive ecosystem research, in addition to adaptive ecosystem management under uncertainty [72]. A long-time perspective on systemic resilience helps to guide conservation strategies that identify vulnerabilities of ecosystems to climate change [73].

Eco-socioeconomic conditions have created numerous vulnerabilities for the human populations in the Amazon, which is worsening the disparity [74]. Overcoming conditions that create vulnerabilities in the bio-chain faces the lock-in mechanisms in the current era of the Anthropocene, where anthropic pressures are placed upon the Amazon region due to the expansion of agriculture and economic development that have profoundly reduced biodiversity, socio-cultural assets and ecosystem services.

To achieve eco-humanistic solutions, the complex character of inequalities in the Brazilian Amazon region needs holistic and systemic interventions that are based on a gender-sensitive perspective with the involvement of all stakeholders based upon pluralism. Taking this into consideration, for interconnecting them to integrate the concept of change within a systemic view of life framework in the efforts for conservation of the biological world, and in building resilient individuals and communities, the authors of this paper elaborated a new approach titled "Resilience in the Systemic View of Life", which envisions that everything and everyone are interconnected.

In order to better feel the worth-of-life/value-of-life, a vision/approach of "Inner Processes of Creation" is discussed, to help those who experience it to become conscious of the worth in him/herself,

and therefore in others and "nature-as-a-whole". The lack of such understanding has led to decisions that have abused human, bio-systems, materials, and financial resources, exclusively for utilitarian purposes, thereby causing disasters and crises for all. This is the case of the huge biome of Amazon, which is under threat and needs urgent care and respect to stop the disasters that are happening and will increasingly occur.

## 4. From Vulnerabilities to Systemic Resilience: Linking Research Questions and Disciplines to Concepts and Measurements

The foregoing sections provided analyses and insights about the Brazilian Amazon region's vulnerabilities and underscored the need for systemic resilience, which is beyond specific contexts. This chapter discusses the need for a systemic resilience and its understanding. The understanding of the resilience concept is important, and measurements can improve our understanding of how people and societies respond to climate risks. It helps to ensure that the effectiveness of resilience-building interventions is tracked over time.

### 4.1. Research Questions

For establishing the discourse on individual and systemic vulnerability and for broadening our perspective, the authors worked through psychoanalytic-linguistic and eco-philosophical perspectives to seek to develop focused questions at different levels:

At the Conceptual Level:

- How is resilience defined?
- How is resilience measured?

At the Individual Level:

- To what extent must the subject (he/she) cope?
- Under which contexts?
- If adversity is an overload on an individual's life, how can the individual overcome this adversity? Alone? By co-working?

At the Political/Ideological Level:

- Are there any hidden denials of fragilities and vulnerabilities in the current discourse of resilience and dominant ideologies?
- Are people/communities/women/biomes at the mercy of "usual and controlling-the-risks strategies", lock-in mechanisms, and colonial/patriarchic dominant ideological patterns of the Anthropocene?
- Are the untold dimensions of adversity the barriers for the continuation of the lock-in mechanisms and anthropocentric-dominant ideology?

At the Eco-Philosophical Level:

- What new paradigm is required for making the needed changes?
- Can the framework of human–nature unity and interconnectedness against the dichotomy of nature–human framework dominant in the Anthropocene bring this shift?
- How can the awareness of the systemic view of life through the awakening of the inner processes of creation lead to decision-making for improved well-being and resilient lives?

We focused upon the individual level because individuals have the following characteristics:

(i) The sovereignty of the "Self".

(ii)    She/He can reason.

(iii)   She/He can use a language.

(iv)   She/He is a subject who is willing to learn a new ideology.

We addressed the following methodological questions:

(a)    How is resilience defined?

(b)    How is resilience evaluated?

A subjective approach was used that included perspectives and judgments of the subject(s) in question, and not the objective ones that can dictate a large degree of our understanding of the processes that shape societal responses to climate change [46]. Adaptation is a complex concept referring to the limits to bio-physical and socio-political, cultural adaptation [49]. Before delving into the nuances of subjectivity and objectivity, it is important to understand what "resilience" means and to review the definitions of resilience and its multifaceted evolution. We opened the resilience discourse by referring to various resilience concepts from multiple disciplines, ranging from psychoanalysis and linguistics to eco-philosophy, which contribute meaningfully to one another towards a transdisciplinary approach.

### 4.2. Disciplines-Definitions

Risk is defined as the probability of loss in terms of human lives, economic assets, environmental sources, species, diversity, cultural values, and critical infrastructure due to an unexpected destructive event that can occur in a certain area in a certain period [75]. Risk is a function of exposure and vulnerability, as depicted in the following mathematical equation [76]:

$$Risk = Hazard*Exposure*Vulnerability.$$

Vulnerability and exposure are dynamic, they vary with time and spatial scales and are economically, socially, geographically, demographically, culturally, institutionally, governmentally, and environmentally interdepended. Individuals and communities are differentially exposed due to differences in wealth, education, social class, religion, gender, age, and health status [77]. Vulnerability is the state of susceptibility to harm from exposure to stresses associated with environmental and social changes and due to the absence of capacity to adapt [78]. It is a function of exposure, sensitivity, and adaptive capacity. It refers to the individual's predisposition to the development of psychopathology or ineffective behavior in situations of crises. It is a multidimensional process affected by social, political, and economic factors interacting, from local to international scales [79]. Gender is a factor of vulnerability related to climate-related impacts [80,81]. There is a broad understanding that gender equality is a fundamental part of increased resilience to disasters [82].

Disaster risk is defined as the possibility of adverse effects in the future. Natural hazards are related to social, poverty, and gender lines. Their livelihood is also more dependent on natural resources that are threatened by climate change. Additionally, they face social, cultural economic and political barriers that severely limit their coping capacities [79].

Mitigation is about reducing the anthropogenic or human causes of climate change, while adaptation concerns adjustments to the consequences for ecosystems and human societies [83]. Coping is the ability to deal with adversity with a set of strategies used by individuals to adapt to adverse or stressful circumstances. It is a key component of the resilience process, a universal human activity, and a universal human experience [84].

### 4.3. Concepts of Resilience

The concept of resilience has gained importance in recent years in the efforts to adapt to climate change and to reduce disaster risks. Resiliency theory has been researched across many disciplines, therefore there is a wide variety of concepts. The term resilience emerged in physics and engineering, by the English scientist Thomas Young (13 June 1773–10 May 1829), who introduced the concept of

elasticity. Additionally, in the field of psychology, the term "resilience" was explored as a cognitive behavioral effort [85].

The concept of resilience was divided based upon perspectives of different "generations" of researchers [85]:

- The first generation defined resilience as "the adaptability of the individual, who can handle and overcome adversity". Based upon that, "Positive Psychology" emerged.
- The second generation added the term "positive adjustment", it encouraged individuals to be stronger and more productive, as it favored the development of the potential of individuals to make them stronger and more productive.

In the field of developmental psychopathology, it was referred to as the ability to cope with challenges and threats while maintaining an internal and integrated sense of self [86]. Neurolinguistics considered that there is no failure, there is only feedback, systems are self-organizing and seek a state of balance and stability [87].

Spiritual resilience (SR) is the ability to sustain one's sense of self and purpose through a set of beliefs, principles, or values while encountering adversity, stress, and trauma by using internal and external spiritual resources. Positive religious coping has a strong relationship with positive adjustment after crises [88].

Ecological resilience (ER) is defined as the persistence of relationships within a system and the ability of these systems to absorb changes and to return to an equilibrium state after a temporary disturbance [89]. In a system profoundly affected by external changes, and continually confronted by the unexpected, questions of loss of existence are raised [89].

Resilience to climate change disasters of individuals, communities, organizations, and countries refers to the adaptation and recovering from hazards, and it starts with disaster risk reduction. The concept is widely coupled with the social-ecological systems and their resistance to climate-change adversities and is gaining attention in discussions on the adaptation theory and practice [90]. Resilience is defined as "the adaptability of the individual, who can handle and overcome adversity" in positive psychology [91]. Given those definitions, resilience is seen as the shared, social capacity to anticipate, resist, absorb, and recover from an adverse or disturbing event or process through adaptive and innovative processes of change, entrepreneurship, learning and increased competence [92].

Equitable resilience (ER) identifies critical issues for engaging with equity in resilience practice. It was defined as a form of human-environmental resilience, which considers dimensions of social vulnerability and differentiated access to power, knowledge, and resources and it takes into account people's perceptions of their positions within their human-environmental system, and for the need of a change to avoid imbalances of power in the future. This definition is embedded in subjectivity, inclusion, scale and transformation and highlights that there are significant interconnections and dependencies among subjectivities, place, identity, and social contexts [93].

During past decades, a new systemic conception of life has emerged at the forefront of science with an emphasis on complexity, change, uncertainty, and self-contained ecosystems, in contrast with the traditional discussion based upon the nature–human systems dichotomy, which makes clear that there is a need for awareness of ecological systems–human interconnectedness [94]. Resilience was defined as a "unifying concept in both ecological and social systems" [95]. Capra and Luigi Luisi (2014) explored "systemic thinking", the implications of the "Systems View of Life" for the global ecological and economic crises and the extent of the adaptation throughout a transformative change [94]. Kuenkel (2017) highlighted the need for transformative changes for SDGs implementation based upon a patterned approach to systemic change that enhances aliveness in socio-ecological systems by supporting the idea that the understanding of such an approach can create a shift from the unsustainable patterns that the current complex and wicked global challenges caused to a more life-enhancing functional one [96]. Syrgiannis et al., (2019) elaborated the concept of "Inner Processes of Creation" by emphasizing that there is a need for the individual to tap into his/her inner wisdom to feel the unity and oneness of her/his relations with the others and with nature [17].

*4.4. Measuring Resilience*

People are resilient to different levels and degrees, depending on the size and type of risk, on coping ability and where they rely. There are weaknesses and dangers in current attempts to quantify resilience. Measuring resilience is being undertaken by academics, non-governmental organizations (NGOs), UN organizations, and national and international inter-organizational panels, including various models from econometric equations and participatory approaches to complex statistical treatment. Indicators are being developed to help to establish standards and to measure and quantify resilience. However, it seems that there are difficulties for measuring resilience because it is not a single ability [97]. Norwegian research has proposed, instead of resilience measuring, the assessing of the vulnerability by using vulnerability metrics to identify potential low-risk (high-resilience, low-vulnerability) and high-risk (low-resilience, high-vulnerability) cases [98]. Other researchers have suggested that vulnerability assessment should emphasize the combination of physical, social, economic and gender characteristics of the region [99,100].

## 5. Why Do We Need a Resilience Approach within the Systemic View of Life Framework?

In this section, the authors used a resilience approach for integrating the systemic view of life, consciousness through the awareness of inner processes of creation, and gender-differentiated vulnerabilities. This framework can be used to significantly contribute to envisioning and implementing new approaches that can cause paradigm shifts that result in transformative changes beyond the SDGs.

*5.1. Research Questions*

The following questions were addressed:

(a)　Resilience of whom to what?
(b)　To what extent must she/he cope?
(c)　Under what circumstances?
(d)　If adversity is an overload on individual's lives, how can this individual overcome this adversity? Alone? With help?

Pelling and Dill (2009) highlighted that the field of disasters is, in fact, a highly political domain [101]. In this section, we addressed the urgency of coping with climate change disasters through philosophical, psychoanalytical and political lens.

The lumped together questions also addressed the following:

(a)　Are there any hidden denials of fragilities and vulnerabilities in the current discourse of resilience?
(b)　Are the untold details of adversity the prime causes for the continuation of the lock-in mechanisms and for anthropocentric dominant ideologies?
(c)　Are we at the mercy of being the ideal subject of the dominant ideology of the Anthropocene?
(d)　Are people/communities/biomes at the mercy of "usual and controlling strategies", lock-in mechanisms and colonial/patriarchic dominant ideological patterns of the Anthropocene?
(e)　How do we analyze transformational socio-ecological capacities and stimulate transitions towards resilient and sustainable societal patterns and norms?
(f)　What new paradigm is required to help to catalyze the urgently needed changes?
(g)　Can the framework of human–nature unity and interconnectedness against the dichotomy of nature–human dominant in the Anthropocene help to catalyze the needed paradigm shift?
(h)　Can awareness of the implications of the systemic view of life through the awakening of the inner processes of creation lead to proper decision-making and result in well-being and a resilient life?

*5.2. Looking at Systemic Resilience through Psychoanalytical and Political Lenses*

In this discourse, the resilience of the individual transcends objective and subjective assessments. We considered an individual who, according to Freud (1923), has the sovereignty of the "self",

his conscience and reason is structured within and by the field of language, but the individual is not always capable of handling or overcoming all adversities. Freud (1990) went beyond the notion of the individual, as he described the unconscious where the "Self" emerges as part of the unconscious mind, influencing the external world, and serving as a mediator [102].

Lacan (1999) re-elaborated the constitution of the "Self" and argued that the "Self" is the subject of language, determined by the representations, traces of memory and signs of perceptions organized in a structure. In this perspective, language reveals the contents of the unconscious [103]. To be a "Subject", it is necessary to inhabit a language system that reveals the unconscious. Lacan (1999) argued that the individual becomes a subject when the individual is subjected to the discourse (language) and communication and, thus, when analyzing the materiality of resilience we can reflect on the presence of an ideology [104].

In the dominant discourses of resilience, often the reference to human fragilities is absent. The suffering is not mentioned because coping brings suffering. A discourse/communication that regulates the relationships between what was said and the unspoken is absent. The non-said relates to the non-saying, that is, in the sense of being implicit (what is not verbalized but is still there). The untold keep secrets between the lines. A discourse that is not referring to the individual's suffering keeps secrets between the lines. Thus, behind speech there is an ideology, that is, a set of ideas, thoughts, doctrines, or world views of an individual or group, for their social and political actions. Resilience is a highly political domain [101]. Therefore, in the dominant ideologies, nothing is said about coping. It is hypothesised that the individual alone has the responsibility for adapting, regardless of the size and type of the adversity. It is expected, no matter what, that he/she must adapt to any adversity [104].

This insight causes us to reflect upon the following:

(a) If adversity is evaluated as an overload, how can an individual overcome the adversity without the support of the community, region and state?

(b) Does the "untold" of adversity qualify as the ideal subject for the dominant ideologies?

We argue that the materiality of resilience reveals the subject's alienation and the ideological subjection [69]. The individual who is expected to adapt alone, without support from the community, region, state, is a person who is lost in the process of being subjected when they are submitted to the discourse of overcoming the adversity. Therefore, she/he becomes the "Object" in the discourse of the "Other" [105]. For these ideologies, no matter what, the individual must adapt to adversity alone [72], she/he is responsible for adapting to adversities, alone [106].

## 5.3. Are We at the Mercy of Being the Ideal Subject of the Dominant Ideology of the Anthopocene?

Each subject has an ideology. For Althusser (1971), interpellation is the process by which ideology addresses the pre-ideological individual and produces him/her as a "Subject" proper of ideology. Ideology is part, or rather, is the condition for the constitution of the "Subject". The interpellation produces the subjection, and this occurs at any historical time and in whatever conditions of production. Therefore, adaptation, coping, overcoming and strengthening are dimensions of ideologies [105].

In the current socio-political paradigm, resilience integrated with the capitalist production envisions the creation of strong and productive individuals, transforming work into an activity linked to strength and productivity, excluding the "will" and conscience of the "Subject". Suffering is excluded also [106]. The industrial society has constructed the productive power that is correlated to the productive "Subject", not only the worker but also the individual who produces well-being, pleasure, and happiness. In this way, one can reflect that in the society in which we live, materiality is the epicenter instead of life. Therefore, it is a "si ne qua non" condition to understand the subject who suffers in the process of seeking to overcome the adversity.

In the Anthropocene we are living in, the "homo industrialis heats, beats, and treats" the Earth for the extraction of resources, making the materiality the essence of existence instead life [107]. Therefore, it is evident that the ideal subject is determined by the productive powers. The understanding of

this aspect reveals that the "subject" becomes an epiphenomenon of social relations, without any interference with his subjectivity, adversities are taken as inevitable.

## 5.4. Looking to Systemic Resilience through Eco-Philosophical Lenses

Climate change is fundamentally an ethical issue. Ethical principles are necessary to combat climate change more effectively. The ethical principle of equity and justice should push states and other actors to encourage public awareness, and participation in decision-making and actions by providing access to information and knowledge [108]. The ethical dimensions of our choices for growth and development lie at the core of such transformative changes according to UNESCO's Declaration of Ethical Principles about Climate Change.

According to Blok (2015), awareness of the ecological crisis is not the problem we face today. The problem is the gap between our ethical judgments about the ecological crises and our ethical responses according to these judgments, which can only be bridged by the enhancement of knowledge and through rational choices [109].

Ethics is important because it is a philosophical approach that can help societies develop a broader view of what are the causes and consequences of climate change and how to react and act accordingly [108]. Gardiner and Hartzell-Nichols (2012) [110] described climate change as a "perfect moral storm" for three reasons:

(a)   Its dimensions: it is a truly global phenomenon.
(b)   Its time: it has intergenerational effects.
(c)   Our incapacities: our theoretical tools are underdeveloped (international justice, intergenerational ethics, scientific uncertainty).

If humans remain centered in anthropocentric behavior to master and exploit the natural world as a commodity for human needs, a fundamental dualism between humankind and nature is developed. Therefore, the transition to ecology thus is seen as a fundamental comportment of self-esteem, respect, and joining in with a broader connectedness [111,112].

## 5.5. Looking to Resilience through Gender Lenses

Gender is an important mediator of how humans view and interact with their environment [49], influencing the use, knowledge, management, access and control over environmental resources [113]. Exploring the gender dimension of resilience to natural disasters helps leaders, researchers and practitioners working to understand people's different relationships to the environment, access to resources and create capacities aiming at managing risk and building gender-sensitive social norms [114]. Gender-focused public policies for climate change adaptation are needed because with the right policy responses, it is possible to empower women to better address the challenges they face. In general, the use of gender-analysis frameworks can guide actors to consider gender dimensions in all processes of risk management for the Brazilian Amazon community and globally, although, sometimes the gender perspective evaporates during the process, or the use of such frameworks leads to a superficial analysis of the changes around equality. In international-development evaluation, practice is usually based on criteria and the measurement of results through a set of indicators. Most researchers have proposed an assessment of vulnerabilities by creating indicators that emphasize the combination of physical, social, and economic characteristics while neglecting the gender dimension [99,100].

Gender-sensitive indicators are necessary to track how different interventions impact the lives of women and men and assess whether progress is made towards gender equality, although issues of gender inequality, such as the poverty of women and their empowerment, are often difficult to measure [82]. Gender-sensitive indicators help explain manifestations of gender inequality that are often invisible in traditional indicators such as the activity, decision-making participation, practical and strategic gender needs profiles, and also the access and control profile. The most important parameters to be measured are:

(a)   Effectiveness (the extent to which the intervention achieved its objectives).

(b)   Efficiency (the degree to which gender equality results are achieved at a reasonable cost).

(c)   Relevance (the extent to which the intervention attends to the different problems and needs of women and men).

(d)   Impact (intervention to a broader policy on gender equality).

(e)   Sustainability (inclusion of strategic gender needs in the intervention).

Along with the critical factors of climate change mitigation, adaptation, technology transfer and financing, it is also important to identify gender-sensitive strategies to respond to the environmental and humanitarian crises caused by climate change, towards sustainability and resilience [115]. Preserving the environment, natural and social resources in the Amazon rainforest, will require a strong government commitment and their willingness to prioritize environmental conservation and the eco-social system's well being within economic development policies [116]. At all levels of leadership and across all sectors of society, women's representation is a necessity [117].

## 6. Proposing an Approach/Vision for Systemic Resilience and Transformation by Integrating the Systems View of Life Knowledge and Consciousness of the Worth/Value of Life

Capra (1989) used the phrase "Uncommon Wisdom" as an extraordinary reservoir of power, love, and wisdom within us [118]. This wisdom can come from inside with introspection and the inner processes of creation can be activated by practices. The inner processes of creation concept is a pathway of awareness of the eco-sphere and human-sphere on planet-earth, in correlation to our responsibilities. Workshops can catalyze this source of knowledge by helping participants to connect with their inner processes of creation to bring light upon what was hidden, thereby revealing it. What was subconscious or unconscious becomes conscious.

The inner processes of creation workshop that activates the awareness of the inner values is focused on the "Subject", rather than the "Object", which enables a deeper understanding of the different issues, particularly regenerative sustainability. The inner processes of creation are an impetus for engaging in sustainable actions, as the vision of change shifts focus from the object to the subject, bringing awareness of how living beings can and must co-work, which is essential for sustainability and resilience. Decisions will then be made by people according to the systems view of life, because people have become aware of their inner worth. With appropriate introspection, our awareness may become effective wisdom to help us as individuals and as members of society to make the needed changes to make eco-social sensitive decisions that are of paramount importance in achieving truly resilient and sustainable eco-social systems [17].

Transformative learning deserves a bigger role in life-long integrated and holistic education as awareness must be interrelated with technological achievements, inventions, and innovations. The process of learning from ourselves is the nearest realm within our reach. This process can be activated by practices, it can be shared in groups working with interdisciplinary and transdisciplinary ideas and approaches. Interdisciplinarity is understood not only as a connection of fields of knowledge, but to regard the individual as being full of possibilities [119].

The dialogics of Morin (2014) consider that interdisciplinarity can emerge even if it appears one discipline is in opposition to other [120]. Transdisciplinarity considers science as focused on the subject, not the object. The object emanates from the subject and both are immersed in a third hidden element of a ratio of infinite reach according to Nicolescu (2014) while oneness can be felt [121].

Nature-centered spiritual traditions and philosophies have many lessons concerning relationships between humankind and nature [122]. The understanding of the connection between our consciousness, thoughts, and actions, and their impact on the world can bring a long-lasting change in any social or environmental system starts. The current environmental crises are, therefore, a clear call to transform our awareness and lifestyles [123] via interdisciplinarity—understood not only as a connection of fields of knowledge but also the way to regard the individual as full of possibilities [124].

In a world of accelerated changes, the ability to seek for the inner wisdom to make decisions is essential. Decision-making affects society, business, and personal lives. Through generating words and correlations, something significant and meaningful becomes conscious, at the individual level and at the group level. Groups and companies that use consciousness as a foundational philosophy can create a more engaging and meaningful customer experience and create emotional connections with their employees and suppliers in their supply chain. They can establish a shared identity based on a clear purposes and values.

We propose an approach to resilience that can be called "Resilience in the Systemic View of Life", which emphasizes that everything and everyone is interconnected. In order to better feel the value of life, consciousness of our "Inner Processes of Creation" activated by practice can help each one who experiences it to become conscious of worth in him/herself and its role in life with others and with nature. Our vision is to help to make people, globally, aware that limits in coping with climate change impacts can be broadened with the "Systems View of Life" framework as a reference to preserve nature, the awareness of "the Worth of Life Within/Without" to recognize the worth of life, and the "Gender Differentiated Vulnerability" as a social epistemological approach to empower communities. With this approach, we can feel our responsibility in helping societies to live within planetary boundaries and be more resilient to disasters and learn that crises can be turned from dangers to opportunities, by using the wisdom from within. This visionary approach considers the individual from the perspective of interdisciplinarity, complexity, and transdisciplinarity, with the capacity of self-organization as an autopoietic being and transformational learning, which is the expansion of consciousness through the transformation of a worldview and specific capacities of the self, facilitated through consciously directed processes. The lack of such understanding has led to decisions that have utilized human, ecological, ethical, material and financial resources exclusively for personal utilitarian purposes, bringing destruction to all of them, and creating the crises we all must face at this time. This is the case of the disrupted Amazon biome, which is under threat and needs urgent care and respect to stop disasters in the chain.

In our approach to resilience, we are critical about the idea promoted by the neo-liberal ideology that everyone can be equally resilient. Vulnerability is not always bad. Human relations are possible with a degree of "opening-up" [91]. What we envision is an individual who can cope with and overcome the adversity or be strengthened in a positive way by facing the adversity. An individual who not only adapts but adapts positively, an individual who moves by the desire of completeness [71]. We expect that individual to be capable of boundless freedom and unfailing submission. A human subject that transcends his/her true state of diffusion and finds an image of himself/herself through language/dialogue/communication. This free subject, without limits, and without flaws reaches social expectations. This definition of resilience considers an individual who does not present signs of discomfort through the subjection of the dialogue with the other. This vision envisions resistant individuals who are following strategies where nothing is said about adversity because the responsibility of adaptation is put on the "Subject".

Figure 1 depicts our vision/approach of "Resilience in the Systemic View of Life" that involves consciousness of the "Value of Life", which is needed for eco-social sensitive decisions and proposals that are of paramount importance for transforming societies into systems, based upon equity, harmony, sustainability and resilience.

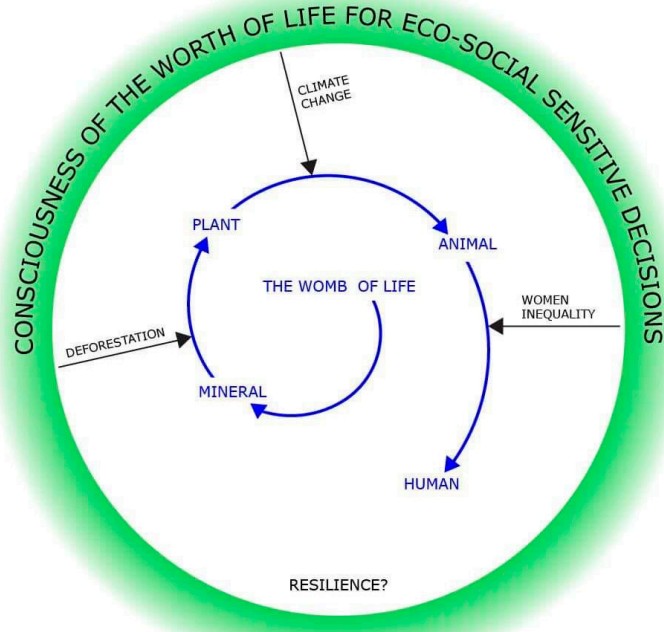

**Figure 1.** Resilience in the systemic view of life: consciousness of the worth of life for eco-social sensitive decisions that are of paramount importance for equity, sustainability, resilience, and healthy, dynamic eco-systems.

We considered resilience as a bridge that spans different disciplines in pursuit of truly sustainable, equitable societies built upon vibrant and dynamic eco-systems. Resilience discourse has the potential to help to bridge different disciplines, by stimulating dialogues among specialists in the natural and social sciences, and among those focused upon science and governance. We stressed the importance of systems thinking, understanding, and accepting anthropogenic climate change-based natural disasters, by focusing upon creative engagement of individuals and societies. As part of a roadmap towards resilience, we particularly emphasized the integration of the recognition of our ethical interconnectedness with others and nature, as a central dimension of the eco-philosophy of life. Introspection can play an important role in thinking and acting to solve societal challenges pertaining to equity, sustainability, and resilience. It was envisioned that with appropriate introspection, awareness can be transformed into effective wisdom and action to help individuals as members of society to make the needed changes to achieve truly sustainable and resilient societal systems. The process to learn from ourselves is the nearest realm within our reach.

Our conceptual framework of resilience in the systemic view of life is depicted in Figure 2.

Figure 2 is a visual representation of the authors' perceptions about moving from multidisciplinarity to interdisciplinarity and transdisciplinary cooperation, regarding each of us being full of possibilities. When we look at how life unfolds in living beings, we feel their wholeness, autonomy, independence, and total interdependence, leading us to fully agree with the systemic view of life, developed by F. Capra [118]. When disruptions in this unfolding system are triggered in the processes, then resilience is required beyond the natural extent. That can be fostered through transdisciplinary visions and approaches by focusing on shifting from the "Object" to the "Subject".

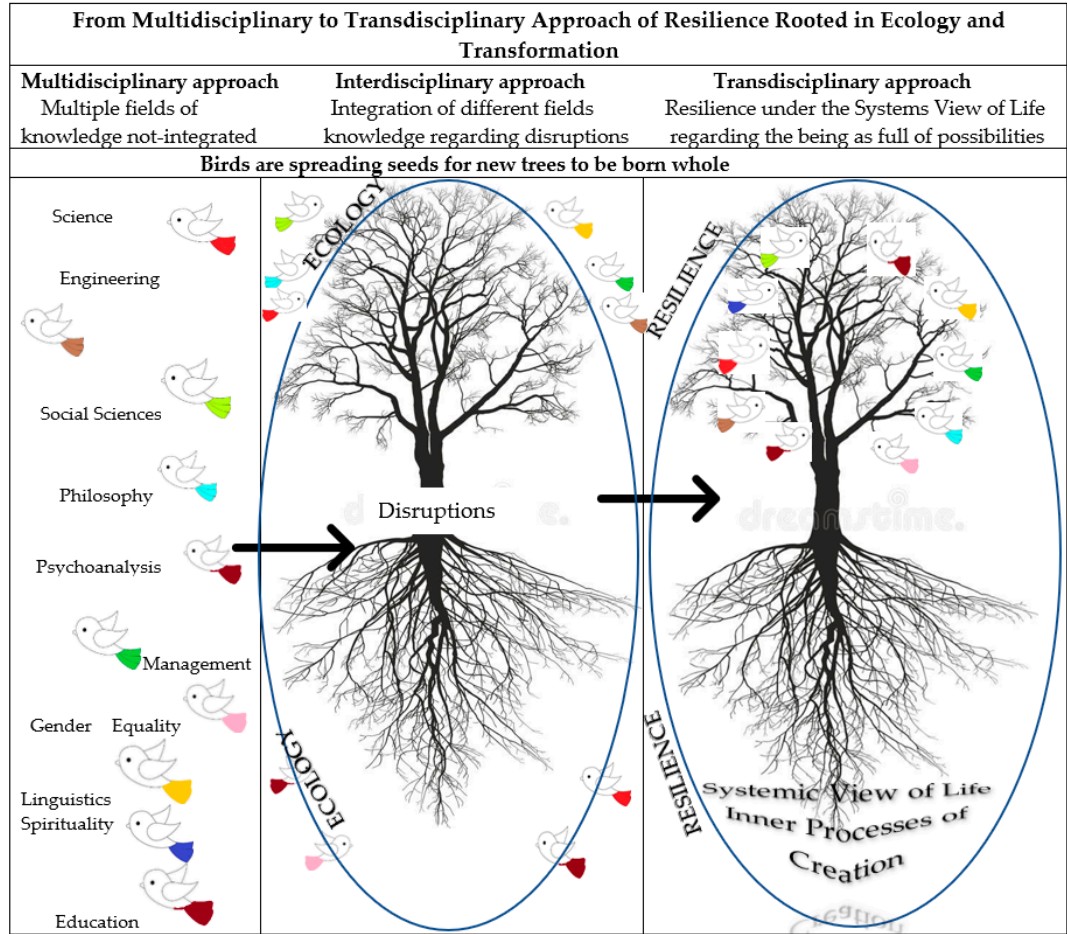

**Figure 2.** From multidisciplinary to transdisciplinary approach of resilience through the systemic view of life and awareness of worth through the inner processes of creation.

## 7. Outcomes

### 7.1. The Genesis of This Paper

The idea for this article emerged during a workshop titled "Inner Processes of Creation", which was held at the "Global Conference on Cleaner Production and Sustainable Consumption", in Barcelona, Spain, in November 2015, with further insights based upon reactions to a presentation of the topic at the "ARTEM Conference on Creativity and Sustainability", held in Nancy, France, in September 2017. The workshop was designed to catalyze the development of perceptions and open the consciousness of the participant's inner wisdom, as it emerges in self-expression, by focusing upon the processes of awakening the wisdom from within, which can help humans to achieve a better understanding of their responsibilities for helping societies to live sustainably within planetary boundaries. The workshop was organized by a Brazilian co-author from the education discipline. Later, a similar workshop was organized and guided by the co-authors in Brazil for academic purposes. When effectively facilitated, this workshop-based process can help participants to gain a deep understanding of the meaning of different issues and to provide support for effective decision-making towards constructive change-making through sustainable proposals made individually and collectively [17].

### 7.2. Moving from Multidisciplinarity to Inderdisciplinarity and Transdisciplinarity

A transdisciplinary approach to systemic resilience triggered the deep interest of the Brazilian co-authors' team that includes scholars from the management discipline, scholars from studies and

research on futures, and from the group of studies and research on interdisciplinarity and spirituality. Further collaboration of those scholars with a Greek scholar from the chemical engineering discipline who works in issues related to sustainable development and gender equality, and with an American professor from environmental studies and sustainability brought out the resilience discourse that can help to develop common understanding by drawing on the differentiation of vulnerabilities/inequalities and building upon the three frameworks (systemic view of life, inner processes of creation and gender equality) by tracing this idea from the beginning to the end as one conceptual argument. The result is a compilation based upon the co-authors' disciplinary engagement and their philosophical beliefs. The first multidisciplinary collaborative output was presented at the International Conference on Innovation and Management (CIM2019), in Jaén, Spain, 26–28 November 2019.

The disciplinary diversity represented in the research involved engineering and technology, management, ecology, sustainability, climatology, education, philosophy, psychoanalysis, linguistics, gender equality and ethics. The authors developed new insights into the system's dynamics of enhancing sustainable, societal resilience by engaging in discourse pertaining to the social, ecological, ethical, economic and gender dimensions of the Brazilian Amazon region's vulnerabilities and inequities. The authors worked to develop an epistemology for transdisciplinary research designed to help to solve "real-world" problems by using eco-ethical-philosophical perspectives instead of using "the 'engineering paradigm" (technological approaches).

Currently, most environmental scientists deal with the complexity of eco-social systems and generate applicable results by adopting concepts and methods from the engineering sciences that have the epistemic objective of producing useful knowledge for solving problems external to scientific practice. This is interpreted by philosophers of science as the emergence of an "engineering paradigm" [125]. Interdisciplinarity is acknowledged in science-based problem-solving research (engineering) and transdisciplinarity is acknowledged in resilience research [126]. Awareness and knowledge can be accessed both from outside and inside.

The authors moved from multidisciplinary to transdisciplinary perspectives of the meaning and urgency of enhancing societal resilience's new conceptual framework. In an interdisciplinary approach, scientists in different disciplines may work together on the same issue, but each can work with her/his own methodologies, which results in outputs and sharing of each other's findings, thus the disciplines become more integrated. In a transdisciplinary approach, stakeholders of various disciplines are required to collaboratively find solutions beyond the limit of single disciplinary knowledge, and to work outside of their own disciplines with the objective of creating sustainable, solution-oriented solutions for complex problems that cannot be solved by a single discipline [127,128]. These solutions can help to make eco-socially sensitive decisions, which are of paramount societal importance for the short and long-term future.

The authors of this paper, having backgrounds in diverse scientific disciplines, and being from different territorial origins, explored dynamic behaviors of eco-social systems in the context of the urgent need to make dramatic changes, and built an interdisciplinary approach to resilience that emerged from their multidisciplinary collaboration and common philosophical beliefs. The result of the joint efforts is an interdisciplinary/transdisciplinary approach to systemic resilience that can be applied to different contexts.

To frame the framework, the authors explored diverse vulnerabilities that revealed the need to seek answers to some relevant philosophical questions, at least partially, and to use diverse lenses such as psychological, political, ecological, technological, and ethical dimensions to seek answers.

In our interdisciplinary discourse/reflections on resilience, we perceived an individual immersed in the resistance to overcome adversity, and we investigated some fundamental issues that have received less attention thus far. We proceeded by looking through psychoanalytical, philosophical, linguistic, political, ethical lenses. We proposed an approach that can become universally applicable. An approach that takes into account the importance of human subjectivity and the variability of human

contexts, especially gender, in shaping human experiences and responses to climate impacts, which is an important set of challenges regarding the inadequacy of current models of resilience analysis.

### 7.3. Proposing an Approach/Vision Not a Model

There is no single, clear framework of resilience that can be applied universally. We proposed an approach, not a model. It is an approach based upon the importance of human subjectivity and the variability of human contexts, especially gender, in shaping human experiences and responses to climate impacts and other types of crises such as the covid-19 pandemic. Our assumptions are that humans are undifferentiated from nature and that complex systems should work reliably in their interactions with humans and natural systems. In our discourse, resilience was an emergent individual experience transcending objective and subjective assessments. The proposed approach/vision can be used to help catalyze needed transformations in human thinking about resilience, sustainability, and life. It is true that these assumptions are being challenged by policymakers and many scientists. However, this is exactly our aim—to challenge them.

We also tried to bridge diverse disciplines to generate a constructive debate on gender-differentiated vulnerability and to embrace the intertwined interactions between resilience and gender into positive, constructive responses. We sought to engage engineering, technology and environmental audiences that usually do not use gender specific lenses in their innovations. The authors envison that this path will result in gender-sensitive proposals made individually and collectively, towards facing the intersectional characritics of gender and power relations in building resilience. Core insights from the systemic view of life and awareness of inner processes of creation can significantly help to advance new sensitive approaches towards transformative changes for enhancing and integrating resilience and sustainability proposals with equality.

### 7.4. From Local to Global

Reflecting on the local scale and bringing together different constructs, the authors opened an eco-philosophical discourse on resilience that is scalable to the global dynamics. They looked beyond the easy answers to be able to create new visions and approaches by using the key perceptions of the systems view of life, and the interdisciplinary principle of regarding the "being" as full of possibilities. Life involves power, love, and wisdom, which can be felt within and can be evidenced outwardly by our actions in society.

Decisions by professionals, authorities, policymakers and by each human being should be valued in this dynamic process. As an alternative to traditional approaches that put emphasis on taking actions designed to adapt a system to given circumstances by providing practical guidance faced with immediate problems, we provided arguments for valuing the worth of life within, based upon the oneness between nature and humans.

### 7.5. Emphasis upon the Need of Humanistic Care and Transformation

Comprehensive humanized care systems require initiatives to strengthen our consciousness, responsibility, and commitment to take care of our own home, with accelerated and converging advances to implement global impact projects within the new sustainability paradigm. These are conditions under which new actions can effectively emerge and be used to support social reforms and to shift away from the negative consequences of human agency in the Anthropocene to the positive, caring and responsible relationships between nature and humans.

The authors of this paper emphasized that there is a need for innovative humanistic approaches to improve individuals' and communities' resilience, to address societal challenges and to pave the way towards climate-resilient equitable, eco-social systems. Knowledge of the natural world should not only be confined to technological solutions. It should encompass the human societies in the context of eco-systems, upon which we are all totally dependent.

Some people have system-based knowledge that includes deep understanding of how we need to work to help to ensure that humans can and do work to avoid catastrophic climate changes. This knowledge and consciousness of nature–human unity and connectedness should be the cornerstone for unity, equality, and wisdom to cope with environmental risks and disasters. Overall, life involves power, love and wisdom, which can be felt within, with the perception of rhythms.

Through outlining these contexts, we have concluded that a gradual transformation of eco-social system structures (including ways of thinking, lifestyles) must be stimulated to achieve resilience and equality. The key to this challenge hinges on fostering awareness of the inner processes of creation and systemic view of life and achieving gender equality, coherence among specific resiliencies so that the sustainability goals of both the ecological and social dimensions will be realized at local and global scales. Our approach, where the human subject transcends the state of diffusion and finds an image of himself/herself, is a transformative change that can enhance societal individual and systemic resilience with and beyond the SDGs. Experiencing the workshop of inner processes of creation is part of the framework and the process for shifting from the "object" to the "subject", helps people to align their thoughts, emotions, words (theory) with actions for life (praxis).

### 7.6. Reducing Fuzziness

We tried to reduce fuzziness and to propose a path that can helps individuals or groups to actively engage in effecting changes based, in part, upon lessons that can be learned via the systemic view of life through the awareness of inner processes of creation processes. The proposed foundation can help people individually and collectively overcome social and territorial assumptions, foster systemic resilience that requires a holistic and integrated approach to implementing the SDGs, respecting human rights and achieving social equality.

This call for systemic resilience involves avoiding monopolistic control of the provision of services. We are critical about the idea promoted by the neo-liberal ideology that everyone can be equally resilient. The claim for ecological consciousness and gender equality in the provision of services should avoid social poverty traps, and unsustainable lock-in mechanisms (learning effects, economies of scale, economies of scope, technological interrelatedness, collective actions, institutional learning effects and the differentiation of power). The rigidity of the functioning of the system and the lack of awareness can create or perpetuate vulnerabilities.

## 8. Recommendations

### 8.1. Future Planning in the Brazilian Amazon

We have attempted to include an example in this study, which was focused upon the Brazilian Amazon biome and gender resilience. The Amazon Forest biome, which is under serious threat, needs urgent care and respect to prevent a catastrophic chain of disasters.

We include arguments and recommendations for future planning approaches for the Brazilian Amazon region's sustainability and resilience:

(1) Transformations can be focused upon three challenges: (a) reducing deforestation and resource consumption, (b) integrating social and environmental criteria alongside economic interests in decision-making, (c) empowering women, mitigating the impacts, and adapting to climate change.

(2) Current patterns of deforestation generate an inherent and naturally inefficient metabolism, which causes extensive and rapid biodiversity losses. The cross-scale impacts of this metabolic behavior hinder sustainable development and resilience.

(3) Workers with native communities in Brazilian Amazon region need to acknowledge gender-differentiated vulnerability of the impacts of climate change and natural hazards and help them to expand their resilience and capacities to act and adapt.

Based upon the findings, the authors emphasize that preservation of the Brazilian Amazon forest and other communities of Amazonas is an urgent case that needs to be faced, not by using the "*controlling-the-risks*" strategy but by adopting the systemic resilience perspective. We suggest that by using insights that can be obtained based upon the systemic view of life and awareness of one's own inner worth, and gender equality, can help in the planning and implementation of the needed changes in the region.

Thus far, the Brazilian Amazon sustainability movement has been regarded as being averse to industrial progress, which is why industrial stakeholders avoid options involving social and ecological sustainability. The 21st century is the century of the indigenous people and of women, because they are actively working to reconnect people to Mother Earth, and to value the ecosystems upon which all humans are totally interdependent. This means it is important to overcome the colonizing power that is mainly masculine. Building disaster-resilient communities beyond the "*controlling-the-risk strategies*" should be the objective of crisis management institutions in the entire Amazon region and elsewhere, globally. Design, implementation, and evaluation of interventions needs to take into consideration the need to anticipate emerging issues with a focus on local/regional dynamics and address them in a constantly changing context. They should take action designed to decrease suffering and to support recovery when a community is faced with adversity.

Although this case study was based upon the Brazilian Amazon forest and communities, we are confident that the approach can be employed in the context of other eco-social challenges. The assertion put forth was that sustainability and resilience follow the precautionary principle regarding resource use and emerging risks, the avoidance of differentiated vulnerabilities and the promotion of ecological integrity and gender equality in the future.

*8.2. Individual's Attitudes*

The authors of this paper wish to recommend attitudes for helping individuals to make progress towards personal and societal resilience:

- Participate with a systemic resilience approach rooted in ecology, diversity, gender equality, societal transformation (actional level) and ethical principles.
- Look at the earth system to understand what is happening (epistemological level).
- Conceive of yourself as having agency, capable of intervening in essential planetary processes at the local and global scale (ontological level).
- Reflect on the fact that exclusion of the subject observed in the discourse of the "controlling the risk" ideology is based upon conviction that one's life is determined and treated as a mere object.
- Defend your subjectivity by giving significance to your identity as part of your life.
- Develop perceptions and consciousness of the inner wisdom, which emerges in self-expression by focusing upon the processes of awakening the wisdom from within. This can lead to a better understanding of your responsibility towards helping societies to live sustainably, to adapt and to be resilient in the contexts of disasters.
- Empower women and participate in gender-sensitive strategies towards decreasing vulnerabilities that are of paramount urgency.

## 9. Conclusions

There are multiple ways of looking at and assessing the resilience of ecosystems, individuals and communities. In this paper, individual's and community's resilience were considered in the context of climate change impacts. A transdisciplinary approach to systemic resilience and transformation with awareness of the "Value-of-Life", human subjectivity, and the gender context to overcome environmental disruptions was used for envisioning new responses to climate change impacts. These visions were used for building discipline-specific constructs and domains, for outlining their interactions, and for interlinking them within the proposal of a systemic resilience approach. The authors recommend

that it is essential to transition beyond the vulnerabilities created by the past and present economic locked-in sociopolitical mechanisms (large-scale resource extraction, unsustainable industrialization patterns, and in the past patriarchical colonization) to the new reality of equitable, systemic, resilience as the path toward more truly sustainable societies.

In envisioning the equitable, systemic, resilience approaches, the authors considered the natural and social capitals, and used the hypothesis that the "Self" is undifferentiated from nature.

Based upon this vision, the authors admit that the proposed approach is not a robust model within a single system, but is a vision of a systemic resilience based upon systemic thinking, transformation, eco-centric ethics, and values as agencies in a complex, integrated system. The value of expanding consciousness through systemic usage of life-based knowledge, embodied practices, and awareness of inner worth/value and wisdom was highlighted. The establishment of an eco-centric orientation of human actions by incorporating the adaptability and self-healing/self-repairing skills to respond to adverse circumstances was envisioned.

The authors hope that the insights presented in this paper may inspire an awakening of the "Subject" in a way that will help people to learn how to align their thoughts, emotions, words and actions for life (theory and praxis), to inspire readers and others to reflect on ways of complementing sustainability science with resilience thinking and to challenge decison-makers to use bold strategies to support more effective and targeted resilience-building interventions locally and globally in the short and long-term future, to help to support the evolution of equitable, sustainable and livable, post-fossil carbon societies.

**Author Contributions:** All authors contributed to several aspects of the study, specifically: conceptualization: C.S., D.G., A.J.d.H.G., I.F. and A.Z.; methodology: A.Z.; formal analysis: A.Z., C.S. and I.F.; investigation, resources, and data curation: C.S. (on philosophical/psychanalytical/political discourse and inner processes of creation, awareness of inner worth of life proposal), D.G. (on linguistic/psychanalytical/political discourse), I.F. (on linguistic/psychanalytical/political discourse), A.J.d.H.G. (on Amazon and political discourse) and A.Z. (on Amazon vulnerabilities, ecological, gender equality discourse, discussion and conclusions); visual representation: A.Z. and C.S.; writing—original draft preparation C.S., D.G., A.J.d.H.G., I.F. and A.Z.; writing the revision: A.Z.; review and editing: D.H.; supervision, A.Z., C.S. and A.J.H.G. Team linking person: A.Z. and C.S. All authors have read and agreed to the published version of the manuscript.

**Funding:** This research received no external funding.

**Conflicts of Interest:** The authors declare no conflict of interest.

## Abbreviations

| | |
|---|---|
| ACM | adaptive collaborative management |
| AS | adaptation services |
| CC | Climate Council |
| CGIAR | Consortium of International Agricultural Research Centers |
| CIFOR | Center for International Forestry Research |
| $CO_2$ | carbon dioxide |
| ER | equitable resilience |
| FAO | Food and Agriculture Organization of the United Nations |
| FLR | forest and landscape restoration |
| GE | gender equality |
| GEC | global environmental change |
| GHGs | greenhouse gas |
| GLP | Global Land Program |
| ILC | International Land Coalition |
| INPE | Brazilian National Spatial Research Institute |
| INPE | Brazilian National Spatial Research Institute |
| MSF | multi-stakeholder forums |
| NbS | nature-based solutions |
| NGOs | non-governmental organizations |
| SCC | systemic change concept |

| SD | sustainable development |
| --- | --- |
| SDGs | sustainable development goals |
| SR | spiritual resilience |
| UN | United Nations |
| UNDP | United Nations Development Program |
| UNFCCC | United Nations Framework Convention on Climate Change |

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
