# Peer review of "From Multidisciplinarity to Transdisciplinarity and from Local to Global Foci: Integrative Approaches to Systemic Resilience Based upon the Value of Life in the Context of Environmental and Gender Vulnerabilities with a Special Focus upon the Brazilian Amazon Biome"

_sustainability, doi:10.3390/su12208407_

Round 1
Reviewer 1 Report
The manuscript entitled “From multidisciplinarity to transdiciplinarity. From local to interrelated global….reflections on the Brazilian Amazon Region” has a great potential to be interesting for the readers of Sustainability journal. It is interesting and innovative review. The following comments and suggestions should be addressed before considering this manuscript for publication.
The abstract should not exceed 200 words. The current abstract includes 394 words.
The aim (objective) of this review is presented in the abstract but is missing in the paper itself. It must be added to the introduction section of the manuscript.
The authors use the phrases “socio-ecological”, social-ecological”, “eco-social” and “socio-environmental” throughout the manuscript several times. Are these phrases interchangeable? If so, please indicate that or use the most common phrase described in the literature (I believe it is socio-ecological or social-ecological).
In the first paragraph of p. 5 the authors present the following three key concepts –“multidisciplinary”, “interdisciplinary” and “transdisciplinary”. I was confused by these concepts and their exact meanings even after reading this paragraph several times. The definitions of these concepts should be presented very clearly. The point of departure of this review paper is moving from multidisciplinary to transdisciplinary approach. The rationale of this idea (of moving from one approach to the other one) should be also described here.
On p. 6 the authors present discourse analysis as their method. It is not clear which data was analyzed through discourse analysis. Are these documents? Specific publications? Interviews with individuals (if yes – please describe with whom?) other sources of data? This point should be better explained.
On p. 6 line 256 the title chosen by the authors is “At general level”. Could it be replaced with “At conceptual level”?
On p. 8 line 317 – I guess it is the third section and not the fourth one.
On p. 14 line 622 it is written: “we reflected upon an individual who has the following characteristics”. What do the authors mean? Did they reflected on a specific individual? Or is it a reflection in the “individual level”? Is this description is where you used the discourse analysis? Please clarify this point.
Regarding Figure 1: By using the word "vegetables" did you mean "plants"?
By using the word "WOMB" did you mean "WORTH"?
Table 1 seems to me as a list of learning and behavioral outcomes. I'm not sure a structure of a Table is needed here.
Looking forward to reading the revised manuscript.
Author Response
General Comment:
It is an interesting and innovative review. The following comments and suggestions should be addressed before considering this manuscript for publication. Looking forward to reading the revised manuscript.
Reply:
We would like to thank you for the very constructive comments.
Comment 1:
The abstract should not exceed 200 words. The current abstract includes 394 words.
Reply 1:
We shortened the abstract, so it contains 200 words
Comment 2:
The aim (objective) of this review is presented in the abstract but is missing in the paper itself. It must be added to the introduction section of the manuscript.
Reply 2:
A new sub-chapter/sub-paragraph (1.1) was added with the scope and objectives of the study.
Comment 3:
The authors use the phrases “socio-ecological”, social-ecological”, “eco-social” and “socio-environmental” throughout the manuscript several times. Are these phrases interchangeable? If so, please indicate that or use the most common phrase described in the literature (I believe it is socio-ecological or social-ecological).
Reply 3:
Thank you for the remark. We now consistently only use the term “socio-ecological” throughout the paper.
Comment 4:
In the first paragraph of p. 5 the authors present the following three key concepts–“multidisciplinary”, “interdisciplinary” “transdisciplinary”. I was confused by these concepts and their exact meanings even after reading this paragraph several times. The definitions of these concepts should be presented very clearly. The point of departure of this review paper is moving from multidisciplinary to transdisciplinary approach. The rationale of this idea (of moving from one approach to the other one) should be also described here.
Reply 4:
The definitions of these concepts and their meaning are presented in paragraph 1.1. In this paragraph, we explain the differences among Multidisciplinarity, Interdisciplinarity, Transdisciplinarity and added the reference 16.
Comment 5:
On p. 6 the authors present discourse analysis as their method. It is not clear which data was analyzed through discourse analysis. Are these documents? Specific publications? Interviews with individuals (if yes – please describe with whom?) other sources of data? This point should be better explained.
Reply 5:
Discourse analysis is described in paragraph 2. Methodology (lines 175-189). Two new references support this analysis (ref 19, 20)
Comment 6:
On p. 6 line 256 the title chosen by the authors is “At general level”. Could it be replaced with “At conceptual level”?
Reply 6:
You corrected to ‘At conceptual level’ as suggested
Comment 7:
On p. 8 lines 317 – I guess it is the third section and not the fourth one.
Reply 7:
Yes, right. We corrected.
Comment 8:
On p. 14 lines 622 its is written: “we reflected upon an individual who has the following characteristics”. What do the authors mean? Did they reflect on a specific individual? Or is it a reflection in the “individual-level”? Is this description is where you used the discourse analysis? Please clarify this point.
Reply 8:
The correct is “individual level where the discourse analysis was used.
Comment 9:
Regarding Figure 1: By using the word "vegetables" did you mean "plants"?
Reply 9:
Figure 1 was corrected and the word ‘vegetables’ was replaced by the word ‘plants’
Comment 10:
By using the word "WOMB" did you mean "WORTH"?
Reply: 10
With the word ‘WOMB,’ we mean “ worth and cosmic mind’
Comment 11:
Table 1 seems to me as a list of learning and behavioural outcomes. I'm not sure a structure of a Table is needed here.
Reply 11:
We change the structure avoiding the table structure. We also numbered this as the sub-paragraph 8.2: Recommendations for Individual Attitudes

Reviewer 2 Report
In my opinion, the manuscript could be improved, my main points are stated below;
- Authors focus too much on the description of the theoretical background and claims what the study will achieve. As the result, only a few sentences present the results of the analysis. This is very little for such an ambitious plan outlined in the text
- I believe one of the weak points of the text is a trial to approach the issue from too many perspectives. Thus, none of the analyses is deep enough and satisfactory. All the parts present a very general outlook on the issue. It is not enough for the research paper.
- line 41: "human applicability"? Is there a possibility of "non-human applicability"?
- lines 42-43 - too broad and unrealistic
- What is meant by a "shared epistemology"? It is impossible for various types of human knowledge to share epistemology. So if authors decide to write about it there needs to be a very profound explanation. All the projects are in the phase of ambitious plans. However, the possibility to create a "shared epistemology" is very dubious.
- If authors would find reasonable claims for "shared epistemology" it should be included in the chapter on methodology
- part of the chapter on methodology is too personal, describe rather how the authors bonded. It would be better to focus on what is relevant to the manuscript
- there is the wrong numbering of the chapters
- part 4.1.3 should start from answering a question: why woman are vulnerable? This issue should open this part of the text
- Equitable resilience (line 660) seems to be a crucial concept, it would be good if authors focus more on this issue
Author Response
General Comment: In my opinion, the manuscript could be improved, my main points are stated below;
Reply: Thank you very much for your effort and time to help us to upgrade the manuscript
Comment 1:
Authors focus too much on the description of the theoretical background and claim what the study will achieve. As the result, only a few sentences present the results of the analysis. This is very little for such an ambitious plan outlined in the text
Reply 1:
Thank you for this very constructive remark. We rearranged the study and many sentences from the Methodology part are placed in the chapter/paragraph 7: Outcomes. This chapter was revised, and it now contains several sub-paragraphs and the recommendations are included in the new chapter 8.
Comment 2:
I believe one of the weak points of the text is a trial to approach the issue from too many perspectives. Thus, none of the analyses is deep enough and satisfactory. All the parts present a very general outlook on the issue. It is not enough for the research paper.
Reply2:
We approached the subject from many perspectives because we wished to use an interdisciplinary approach. The various scientific analyses obviously cannot be very deep due to the limited length of the manuscript. We could say that this manuscript is a mix of a review paper, a case study and discourse analysis. All methodologies are explained in chapter/paragraph 2. We created an approach/vision and not a Model. Models that usually explain resilience within a single system are more robust. Our purpose was not to present a theoretical model but an approach of resilience that depicts our eco-philosophical perspectives that it can result in transformative thinking and acting. The insights of the study may inspire an awakening of the ‘Subject’ in a way to help people to learn how to align their thoughts, emotions, words and actions for Life (Theory and Praxis), to inspire readers to reflect on ways of complementing sustainability science with resilience thinking and decision-makers to develop and use bold, robust strategies to support more effective and targeted resilience-building interventions on the ground, globally in the short and long-term future, for more equitable, sustainable and livable societies.
Comment 3:
line 41: "human applicability"? Is there a possibility of "non-human applicability"?
Reply 3:
We replaced ‘human’ with ‘local and global’ applicability
Comment 4:
lines 42-43 - too broad and unrealistic
Reply 4:
The resilience approach we proposed is an integrated, holistic, subjective and systemic vision; we envisioned that this approach can transform human experiences and responses to climate impacts. Our assumption is that complex systems should work reliably, in their interactions with human and natural systems. It is true that this assumption is being challenged by policymakers and many scientists. However, this is exactly our aim: to challenge them.
Comment 5:
What is meant by a "shared epistemology"? It is impossible for various types of human knowledge to share epistemology. So if authors decide to write about it there needs to be a very profound explanation. All the projects are in the phase of ambitious plans. However, the possibility to create a "shared epistemology" is very dubious. If authors would find reasonable claims for "shared epistemology" it should be included in the chapter on methodology
Reply 5:
We deleted the claim of "shared epistemology."
Comment 6:
Part of the chapter on methodology is too personal, describe rather how the authors bonded. It would be better to focus on what is relevant to the manuscript
Reply 6:
We moved this part from methodology to chapter 6: Outcomes. We added all those personal details because this study was based upon a workshop where co-authors participated in or guided the workshop.
Comment 7:
there is the wrong numbering of the chapters
Reply 7:
Thank you. We corrected.
Comment 8:
part 4.1.3 should start by answering a question: why woman are vulnerable? This issue should open this part of the text ‘why woman are vulnerable?’
Reply 9:
Thank you. We started the part 4.1.3 with the above question
Comment 9:
Equitable resilience (line 660) seems to be a crucial concept, it would be good if authors focus more on this issue
Reply 9:
Thank you. We focused more (lines 656-662) and we also added the ref. 93.

Round 2
Reviewer 1 Report
Well done!
The current version of the manuscript seems ready for publication
This manuscript is a resubmission of an earlier submission. The following is a list of the peer review reports and author responses from that submission.
Round 1
Reviewer 1 Report
This study proposes what appears to be an interesting and important thesis regarding the inadequacy of some current models of resilience analysis, proposing instead a model that takes account of the importance of human subjectivity and the variability of human contexts, especially gender, in shaping human experience of and response to climate impacts. However, several problems mar the paper, impeding comprehension beyond the most rudimentary level and thereby making its actual significance hard to evaluate. For that reason I did not mark a response under the categories "Significance of Content" or "Interest to the Readers" in the chart above -- because it is impossible to gauge these qualitative factors in the paper's current state. A primary problem is the need for extensive English-language editing. Many, many sentences are simply inscrutable; and even in places where the meaning is clear, mistakes and infelicities abound. But even with these problems resolved, the paper fails to make clear fundamental methodological questions such as how and why the study's authors chose the various discourses engaged here (ranging from climate science to psychoanalysis, linguistics to [eco-] philosophy, as well as the study of human resilience) or how exactly these wildly diverse areas of discourse are meant to contribute meaningfully to one another rather than apparently simply being placed alongside one another. It is additionally unclear how if at all the study's conclusions might pertain differently to vastly different populations within the enormous Amazon biome under consideration, let alone to people outside the Amazon. To put the point somewhat differently: who exactly is the "we" assumed in Table 1 (lines 215-216)? For a model emphasizing human particularity and subjectivity, surely there can be no assumption of universal human applicability. I suggest reworking the paper beginning with an actual case study of a particular community of men and women in the Amazon, known intimately enough to be able to show gender variability as well as the community's particular vulnerabilities in the face of climate change -- and then from this case articulating why each of the disciplines used in the analysis emerges necessarily as essential to grasping the full scope of this community's resilience. From there, the authors can show how their proposed interdisciplinary model of resilience more adequately accounts for this community's experience -- particularly the gendered dimensions of experience -- in comparison with alternative models of resilience.
Author Response
We would like to thank you and the other reviewer for your constructive comments.
According to your comments, we restructured, upgraded, clarified our objectives and approach to resilience and advanced our methodology and outcomes' presentations as well as the results. We included two figures to better explain the concept.
Our vision is not to present a model but an inspiration for resilience thinking that takes the systems view of life in consideration and as you say, it takes account of the importance of human subjectivity and the variability of human contexts, especially gender, in shaping the human experience of and response to climate impacts. This became obvious in the new version, in the description of the objectives and the methodology. Figure 1 depicts very well our approach.
As you requested, we presented more clearly the fundamental methodological questions, such as how and why the study's authors chose the various discourses engaged here (ranging from climate science to psychoanalysis, linguistics to [eco-] philosophy, as well as the study of human resilience) or how exactly these wildly diverse areas of discourse are meant to contribute meaningfully to one another rather than apparently simply being placed alongside one another. You can see the clarification all over the paragraph of the methodology included figure 1.
As you suggested, we reworked the paper, beginning with an actual case study of a particular community of men and women in the Brazilian Amazon, and discussed some vulnerabilities (ecological and gender) because those are pertinent to our experience and knowledge. From this case we articulated why each of the disciplines used in the analysis emerges necessarily as essential to grasping the full scope of this community's resilience. From there, the authors can show how their proposed interdisciplinary model of resilience more adequately accounts for this community's experience -- particularly the gendered dimensions of experience -- in comparison with alternative models of resilience.
We took Amazon biome and gender vulnerabilities as a case study to continue the discussion of an individual’s resilience under a philosophical lens. Our purpose was not to go deep in analyzing one parameter or population or issue but to present how our collaboration that started from different disciplines has managed to bridge the gaps through our eco-philosophical beliefs that are common for all co-authors.
We reworked the paper, beginning with an actual case study of a particular community of men and women in the Brazilian Amazon, and discussed some vulnerabilities (ecological and gender) because those are pertinent to our experience and knowledge. From this case, we articulated why each of the disciplines used in the analysis emerges necessarily as essential to grasping the full scope of this community's resilience. From there, the authors can show how their proposed interdisciplinary model of resilience more adequately accounts for this community's experience -- particularly the gendered dimensions of experience -- in comparison with alternative models of resilience.
We corrected the errors of the English language, as possible.

Reviewer 2 Report
There is much that I appreciate about this article. It is sincere. It is expressing views that resonate deeply with my own views. And I would like to see this article published eventually. I believe that the ideas presented here are significant and I hope that the authors can take some time to articulate them in a form that would address some of the following issues.
First, and I realize that none of the authors are native English speakers and I understand how hard it is to write in a foreign language, the prose and writing mechanics needs a lot of work. I trust this can be taken care of relatively easily and I hope the authors can find someone who can help them with the editing.
Second, and more challenging...as I said above, I am sympathetic to much of what the authors are presenting. The biggest problem, in my mind, is that the authors are presenting so many different ideas. It may be that there is a disciplinary practice I am unfamiliar with. As a philosopher who works in environmental humanities, I engage with a lot of the ideas/scholarship/themes being addressed here. However, I am not familiar with treating these ideas in such a cursory fashion. There are so many ideas. And the authors move through the ideas so quickly. For someone like myself, who agrees with much of what the authors write, maybe that is fine. But the authors are not making many arguments to try to persuade anyone who might disagree to take a different position. And where there are points where I do feel like maybe some more nuance might be helpful, there is not enough room, as the authors are speeding on to the next idea.
Again, perhaps this is a disciplinary difference. There was a great deal of attention devoted to setting up the study...up through page 9. I think that it would have made for a more interesting article, and more compelling, to take some of the ideas presented in the study, most importantly, that we need a broadening of the understanding of resilience, and just have the whole paper devoted to that idea.
If it were me, I think I would just try to focus on the idea of resilience, problematize the common understanding by drawing on the ideas of, for example, differentiation of vulnerabilities, etc. and trace this idea from the beginning to the end as one conceptual argument.
Author Response
We would like to thank you and the other reviewer for your appreciation and very constructive comments.
According to your comments, we restructured, upgraded, clarified our objectives and approach to resilience and advanced our methodology and outcomes' presentations as well as the results. We included two figures to better explain the concept.
Our vision is not to present a model but an inspiration for resilience thinking that takes the systems view of life in consideration and as you say, it takes account of the importance of human subjectivity and the variability of human contexts, especially gender, in shaping the human experience of and response to climate impacts. This became obvious in the new version, in the description of the objectives and the methodology. Figure 1 depicts very well our approach.
As you requested, we presented more clearly the fundamental methodological questions, such as how and why the study's authors chose the various discourses engaged here (ranging from climate science to psychoanalysis, linguistics to [eco-] philosophy, as well as the study of human resilience) or how exactly these wildly diverse areas of discourse are meant to contribute meaningfully to one another rather than apparently simply being placed alongside one another. You can see the clarification all over the paragraph of the methodology included figure 1.
As you suggested, we reworked the paper, beginning with an actual case study of a particular community of men and women in the Brazilian Amazon, and discussed some vulnerabilities (ecological and gender) because those are pertinent to our experience and knowledge. From this case we articulated why each of the disciplines used in the analysis emerges necessarily as essential to grasping the full scope of this community's resilience. From there, the authors can show how their proposed interdisciplinary model of resilience more adequately accounts for this community's experience -- particularly the gendered dimensions of experience -- in comparison with alternative models of resilience.
We took Amazon biome and gender vulnerabilities as a case study to trigger the discussion of resilience under a philosophical lens. Our purpose was not to go deep in analyzing one parameter or population or issue, but to present how our collaboration that started from different disciplines managed to bridge the gaps through our eco-philosophical beliefs that are common for all co-authors.
It seems that we presented many different ideas but that was the purpose: how to bridge different disciplines under one concept and build a transdisciplinary approach to a complex problem, It may be that there is a disciplinary practice I am unfamiliar with. We explain this very well in this upgraded version, in paragraph 2 and 3 (objectives and methodology). As we explain in the text, we moved from multidisciplinary to interdisciplinarity and transdisciplinary resilience’s new conceptual framework. The traditionally common disciplinary studies, that are based on a single discipline approach, are conducted mainly within the bounds of a single discipline that works with discipline-specific questions-hypotheses-theories-models-methods, while there is no significant linkage with other disciplines. With an inter-disciplinary approach, scientists in different disciplines may work together on the same issue, but each can work with her/his own methodologies that result in individual outputs and then they may share the same methodologies and each other’s findings, thus the disciplines become more integrated. Instead, the transdisciplinary approach requires different stakeholders of various disciplines to collaboratively find solutions beyond the limit of single disciplinary knowledge, and to work outside of own disciplinary, aiming to create sustainable solution-oriented knowledge for complex problems that cannot be solved by a single discipline helping in taking eco-social sensitive decisions that are of paramount importance.
As also we explain in the methodology, the idea of a transdisciplinary approach to resilience triggered first the Brazilian group that includes scholars from Management discipline, a group of Studies and Research on Futures, and a group of studies and Research on Interdisciplinarity and Spirituality. Further collaboration of those scholars with a Greek scholar from Chemical Engineering discipline, working on sustainable development and gender equality, brought out the resilience discourse that is a compilation out of co-authors’ own disciplinary engagement and of their philosophical beliefs, that can problematize the common understanding by drawing on the differentiation of vulnerabilities/inequalities and building on the three frameworks (Systemic View of Life, Inner Processes of Creation and Gender equality); tracing this idea from the beginning to the end as one conceptual argument. The first multidisciplinary collaborative output was presented at the CIM2019: International Conference on Innovation & Management, at Jaén, Spain, November 26-28, 2019.
We followed your suggestion to focus on the idea of resilience, problematizing the common understanding by drawing on the ideas of differentiation of vulnerabilities, etc. and trace this idea from the beginning to the end as one conceptual argument. We did so in the new version and thank you again for this suggestion. It facilitated the flow of the paper.
We corrected the errors of the English language, as possible.
